# Redox-Regulation in Cancer Stem Cells

**DOI:** 10.3390/biomedicines10102413

**Published:** 2022-09-27

**Authors:** Uwe Lendeckel, Carmen Wolke

**Affiliations:** Institute of Medical Biochemistry and Molecular Biology, University Medicine Greifswald, D-17475 Greifswald, Germany

**Keywords:** antioxidant signaling pathways, cancer stem cell, CD13, drug resistance, Sonic Hedgehog signaling, Nrf2-sinaling, ROS, Wnt-signaling

## Abstract

Cancer stem cells (CSCs) represent a small subset of slowly dividing cells with tumor-initiating ability. They can self-renew and differentiate into all the distinct cell populations within a tumor. CSCs are naturally resistant to chemotherapy or radiotherapy. CSCs, thus, can repopulate a tumor after therapy and are responsible for recurrence of disease. Stemness manifests itself through, among other things, the expression of stem cell markers, the ability to induce sphere formation and tumor growth in vivo, and resistance to chemotherapeutics and irradiation. Stemness is maintained by keeping levels of reactive oxygen species (ROS) low, which is achieved by enhanced activity of antioxidant pathways. Here, cellular sources of ROS, antioxidant pathways employed by CSCs, and underlying mechanisms to overcome resistance are discussed.

## 1. Introduction

In cancer therapy, a significant limitation of many currently used chemotherapeutic protocols, and thus a longer-term remission, is that so-called tumor stem cells (cancer stem cells, CSC) evade the effect of the therapeutic agents. CSC show a progenitor cell-like phenotype, are found as a regular cellular population in most tumors or tumor cell lines, and, due to their ability to self-renew, contribute to tumor initiation, progression, and spread as well as to its resistance, recurrence, and metastasis after therapy [1,2]. Cancer stem cells are a self-renewing subpopulation of tumor cells that, due to their capacity for multilineage differentiation, give rise to all cell types present in a given tumor sample. CSCs evade chemotherapeutic attacks by expressing highly efficient efflux pumps [3]. The rapid extrusion of Hoechst 33342 is an established FACS-based technique with which this CSC (which can be defined and represented in FACS as a so-called “side population”, SP) can be quantified and enriched by “sorting” [4,5]. The CSC phenotype is characterized by, among other things, the expression of various surface markers. A continuous transition between CSC and mature tumor cells as well as CSC and non-tumor cells contributes to the expansion of the tumor mass, and is also regarded as the basis for the indefinite maintenance of tumor cell lines (7). CSC stemness is controlled by multiple signals from CSCs themselves and from the tumor microenvironment, which includes the so-called CSC niche [2]. Among the pathways that regulate both CSC differentiation and non-CSC dedifferentiation to achieve a progenitor phenotype, (non)canonical Wingless-Int (WNT), transforming growth factor/bone morphogenic protein (TGF/BMP), Sonic Hedgehog (HH), and Notch pathways are well established and most relevant signaling pathways [2] (Figure 1). A significant influence on the oncogenic transformation of CSC or the differentiation of CSC from non-malignant stem cells [6] is also attributed to the coagulation system [7,8] and the presence of reactive oxygen species (ROS)/hypoxia [9]. The contribution of the latter to the CSC phenotype and tumor development, maintenance, and progression is the main subject of this review.

## 2. “Side Population” Cells in Cancer

According to the CSC hypothesis, tumors typically consist of a heterologous mixture of tumor cells. Only a small proportion of tumor cells is in fact capable of inducing tumor growth when transferred to immunodeficient mice or, e.g., semi-solid media in vitro. This stem cell-like subpopulation is characterized by high proliferative potential, the capacity to self-renew and rebuild progeny [10]. Various cell surface proteins, including CD133 and CD44, have been identified as cancer stem cell markers [11,12]. Markers that would allow for the identification and isolation from tumors of CSCs are often lacking specificity and mostly rather represent markers for stem cells in general, as is true for CD133, CD44, and the aldehyde dehydrogenase, ALDH [10]. In the absence of tumor entity-specific CSC markers, the rapid extrusion of Hoechst 33342 by ABC transporters has become an established FACS-based technique with which these CSCs (which can be defined and represented in FACS as a so-called “side population”, SP) can be quantified and enriched by “sorting”.

As CSCs present themselves as a “side population” in FACS analysis, all the typical characteristics of CSCs of course apply to SP. Only a few studies directly investigated the effects of ROS on SP. Results of recent studies demonstrated, however, that in SP cells the intracellular concentration of ROS is low and expression of hypoxia inducible factor 1 (HIF1A) was found largely restricted to SP cells, where it promoted the expression of the anti-oxidant, heme oxygenase-1 [13].

In contrast, the exposure of SP cells to hydrogen peroxide induced CSC differentiation. Along this line, in bladder cancer cells, administration of dexamethasone to induce cell differentiation decreased the fraction of “side population” cells concomitant with the induction of intracellular ROS and loss of stemness-associated factors, including SNAIL, OCT4, or β-catenin [14].

The presence of this side population, which exhibits high drug efflux capacity, was first confirmed in human tumors by Goodell et al. [3] and has been applied since to many cancers [15,16,17] or cancer cell lines [18,19,20,21]. A special contribution to the optimization and standardization of the FACS protocol for the quantification of human and murine SP cells was made by Anna Golebiewska and her group [4]. The SP phenotype has been suggested to explain both resistance to chemotherapy [22,23,24,25] as well as higher tumorigenicity when xenografted into immunodeficient mice [26,27,28,29,30]. A population of glioma-inducing cells from human glioma tissue has been enriched on the basis of cellular autofluorescence and morphology only, instead of applying surface markers for selection [31].

## 3. Reactive Oxygen Species and Sources in CSC

Tumors, and CSCs in particular, typically exhibit a very tight regulation of ROS concentrations, which is essential for cell signaling, therapy resistance, and tumor recurrence [9]. Alterations in the cellular redox state can be due to both increased production and reduced removal of ROS [32]. In the various cell compartments, very different ROS species are formed, each of which has a special effect on tumorigenesis, tumor progression, or therapy.

ROS, generally, can be divided into two classes: free radical ROS and nonradical ROS. Whereas the former class includes the members superoxide anion (${\cdot O}_{2}^{-}$), nitric oxide (·NO), and hydroxyl ion (·OH), the latter class includes hydrogen peroxide (H_2_O_2_) and peroxynitrite (^−^ONOO).

Superoxide anion and hydrogen peroxide are certainly the best-characterized ROS [33,34]. Mitochondria are considered the major source of ROS [35]. In the mitochondrial respiration chain, 0.1–1.0% of all consumed oxygen by mitochondria can undergo one-electron reductions to yield ${\cdot O}_{2}^{-}$ [36,37]. The majority of ${\cdot O}_{2}^{-}$ originates from complexes I and III of the electron transport chain [37]. The superoxide anion radical (O^−2^) is subsequently converted into H_2_O_2_ by either superoxide dismutase (SOD) [38] or spontaneous dismutation. Due to its relative stability and lack of charge, H_2_O_2_ may diffuse across longer distances and, thus, can easily attack mitochondrial DNA or diffuse to the nucleus where chromosomal DNA gets damaged.

Another important cellular source of ROS is the family of NADPH oxidases (NOXs). Here, regulation relies on the signal-induced assembly of the NADPH oxidase. Membrane-bound NADPH subunits p22 and gp91 become assembled with the subunits p40, p47, and p67, which are initially located cytosolically, to provide the active NADPH oxidase. The formation of the active NADPH oxidase results in increased oxygen consumption. It has been suggested that the activation of NOXs in cancer cells contributes to oncogene-mediated cancer cell survival [39].

·NO is a short-lived, ubiquitous signaling molecule implicated in a variety of ways in tumorigenesis and progression [40]. NO is formed by constitutively expressed (endothelial NOS/eNOS/NOS3 or neuronal NO synthase/nNOS/NOS1) or inducible isoforms (inducible NOS/iNOS/NOS2) of NO synthase. An increase in the expression/activity of iNOS and the enhanced ∙NO production that results is associated with various pathophysiological conditions, including inflammation, ischemia/hypoxia, and also cancer. Mechanistically, ·NO is formed by the oxidation of L-arginine which yields citrulline in addition to ·NO. ·NO exerts multiple effects on tumor cells, including both pro- and antitumorigenic effects depending on multiple aspects, which include the actual cell type/tissue, the amounts generated, the oxidative/reductive environment, the tumor environment, and surrounding/infiltrating immune cells and their polarization. The diverse roles of ·NO, along with most recent NO-related anticancer therapeutics and applications, have been comprehensively and excellently reviewed recently [40].

That NO is relevant to CSCs at all was first suggested in breast cancer, where NO increased expression of CD44 and of signal transducer and activator of transcription 3 (STAT3) [41]. Accumulating evidence shows that increased NO production by upregulated iNOS or eNOS supports a stem cell-like phenotype in various tumors, including prostate and bladder cancer [42,43], glioma [44,45], and liver and colon cancer [46,47]. Mechanistically, activated Notch signaling [46], increased expression of β-catenin [47], and transcriptional activation or protein stabilization of stem-cell associated transcription factors (SOX2, OCT4) or increased expression of stem cell markers such as CD133 and ALDH1 [45,48,49] contribute in NO-regulation of CSCs [50].

## 4. Reactive Oxygen Species in Tumorigenesis and Disease Progression

Reactive oxygen species (ROS) are generated under physiological conditions where they function as second messengers in numerous redox-sensitive signal transduction pathways. Oxidative stress has been recognized to play a key role in the pathophysiology of cancer. In tumor cells, maintaining a precise redox balance is a lifelong challenge. Depending on the microenvironment, which is formed by surrounding/infiltrating (immune-)cells, cytokines, and cells of the tumor niche, ROS produce two kinds of effects on tumors. Whereas appropriate concentrations of ROS stimulate tumorigenesis and promote disease progression, higher amounts of ROS cause cell death [51]. Solid tumors and leukemic cells have been demonstrated to produce increased amounts of ROS when compared to normal cells [52,53]. Nevertheless, the elevated ROS levels typically observed in tumors appear to be controlled and limited to a tumor-promoting level, which is achieved by means of the strong induction of an antioxidant equipment. However, “oxidative stress” may occur when the increased production of ROS can no longer be neutralized by anti-oxidative mechanisms. In pathophysiological conditions, chronically elevated amounts of ROS cause molecular damage which impairs cellular structures and functions, known as “oxidative stress” [51]. Historically, the term “oxidative stress” was defined as an imbalance between the generation of ROS and the capacity of the ROS-inactivating defense systems that are intended to detoxify them [54]. Some key findings have changed the understanding in recent years: different oxidants affect distinct subsets of target proteins through modifications that are specific both with respect to the oxidant and the site of modification, most frequently well-defined cysteinyl side chains. The antioxidant redox systems in the different cellular compartments, e.g., glutathione, NADPH, thioredoxin (Trx), and peroxidases such as the peroxiredoxins (Prx), are not in equilibrium but independently maintained at distinct redox potentials. Oxidative stress may thus, more contemporarily, be defined as the chronic dysregulation of redox systems and redox-responsive signal transduction pathways [55,56,57].

Similarly to normal stem cells, CSCs contain lower amounts of intracellular ROS than cells of the tumor bulk [58,59], which is mainly due to elevated expression levels of ROS scavengers [59,60,61]. As shown by Dong et al., the zinc finger transcriptional repressor, Snail, which is implicated in epithelial to mesenchymal transition (EMT), can alter cellular metabolism of breast cancer cells to that of CSCs [62]. The reduced expression of fructose-1,6-bisphosphate promoted glycolysis and formation of NADPH via the pentose-phosphate pathway.

Lower levels of ROS in CSCs are not only important in maintaining a stem cell-like phenotype, but also confer resistance to radiation or chemotherapy which is partly due to less DNA damage occurring upon therapy [63]. It could be confirmed in CD13^+^ liver CSCs that the pharmacological inhibition of APN/CD13 makes CSC more sensitive to the genotoxic chemotherapeutic 5′-fluorouracil (5-FU) [63,64]. Notably, inhibition of APN/CD13 suppressed the self-renewing and tumor-initiating activity of CD13^+^ dormant CSC [63].

## 5. Redox-Signaling in CSCs: Role of NRF2

The transcription factor erythroid 2-like 2 (NRF2) is of paramount importance to the defense against oxidative stress by the induction of antioxidant protein, ROS-degrading enzymes, and efflux transporters.

NRF2 plays a special role in CSC maintenance and resistance to cancer therapy protocols. Notably, the low ROS levels which are typically observed in normal stem cells and CSCs and which are required for stem cell maintenance result from Nrf2 signaling [38,65]. Under normal conditions, NRF2 is constantly degraded upon binding to KEAP1 in a ubiquitin-proteasome dependent manner [66,67,68]. When cells encounter oxidative stress, NRF2 is released from Kelch-like ECH associated protein 1 (KEAP1) and translocates into the nucleus, where it activates antioxidant response element (ARE)-containing genes such as the antioxidants heme oxygenase 1 (HMOX1), NAD(P)H quinone oxidase (Nqo1), thioredoxin (TRX), peroxiredoxin 1 (PRDX), glutathione-S-transferases (GSTA1, -2, GSTM1-4), or the efflux pump ATP binding cassette subfamily C member 1 (ABCC1, MRP) [38,69,70] (Figure 2).

Early work identified Nrf2 activators as chemopreventive substances in animal models of chemically induced carcinogenesis and diseases associated with oxidative stress/inflammation [73,74]. Despite being protective for normal cells, NRF2 has stimulating effects on tumor growth/progression and increased resistance to chemotherapy, as has been shown for a variety of human cancers [70,75,76,77]. Persistent activation of Nrf2 signaling has been detected in many different human cancers [70] and is frequently the result of somatic mutations of the *NRF2*/*KEAP1* genes [78]. Hypermethylation of the KEAP1 promoter and resulting downregulation of KEAP1 expression levels represent another relevant mechanism leading to a constitutive NRF2 activation in cancer [79,80,81,82,83].

In CSCs, low ROS levels are a prerequisite for maintaining stemness, which is achieved to a large extent by NRF2 keeping these cells in a stem cell-like and undifferentiated state [84]. Accordingly, downregulation of NRF2 reduced the expression of stem cell markers in various types of cancer cells and provoked cellular differentiation in a glioma model [85,86,87]. Notably, NRF2 has been shown to bind to upstream regions of the genes for stem cell factors, *NANOG HOMEOBOX* and *OCT4* [88].

Breast cancer CSCs show an increased expression of the NRF2 target gene, glutamate-cysteine ligase catalytic subunit (*GCLC*) [89]. GCLC is a rate-limiting enzyme of glutathione synthesis and, thus, adds to the CSCs’ antioxidative equipment.

Exposure to chronic and low levels of arsenic stress induces transformation of the human bronchial epithelial cells, BEAS-2B, to cancer stem-like cells. This dedifferentiation of BEAS-2B cells into CSC-like cells was accompanied by features of EMT and enhanced chemoresistance [90]. Exposure to arsenic also multiplied the skin carcinogenesis in Tg.AC mice, a mouse strain where the activation of the v-Ha-ras transgene increased the induction of 12-O-tetradecanoyl phorbol-13-acetate (TPA)-induced squamous cell carcinomas, along with increasing the number of CD13-positive CSC [91]. A Nrf2-dependent metabolic shift towards glycolysis very likely contributes to this effect of arsenic [92].

NRF2, via induction of its target genes, also induces anticancer drug resistance in CSCs. Increased expression of NRF2 and ABCB1 (MDR1) was associated with increased resistance to doxorubicin of CD44^+^CD133^+^ cells [93]. Likewise, increased expression of Sonic Hedgehog (SHH) and NRF2, correlated with induced CSC characteristics and chemoresistance in head and neck squamous cell carcinoma [94].

Recent work suggests that histamine N-methyltransferase (HNMT), which is regulated by ROS, is co-expressed with human epidermal growth factor receptor (HER2) [95]. The expression of both HER2 and HNMT is increased in breast and lung cancer [95,96]. HNMT directly interacts with HER2 to regulate NRF2 DNA binding activity and target gene expression, which is in part mediated by phosphatidylinositol-4,5-bisphosphate 3-kinase (PI3K) [95]. In non-small-cell lung cancer (NSCLC) CSCs, expression levels of the HNMT-inhibiting miRs, miR3065/223, are low, leading to elevated expression/activity of NRF2. This contributes to enhanced chemoresistance to cisplatin [95].

NRF2 also plays an important role in macrophage polarization and, thus, macrophages are considered key players in cancer progression [75,97,98,99]. Infiltration of macrophages in and around tumors (tumor-associated macrophages, TAMs) can often be observed, and their presence is associated with poor prognosis in solid tumors [100,101]. It is considered proven that M2 macrophages support angiogenesis and neovascularization, remodeling of stroma and the tumor niche, and dissemination of tumor cells. All these processes contribute to tumor progression. Fast-growing solid tumors typically contain hypoxic areas due to insufficient oxygen supply. These areas are often located in the tumor center or in perinecrotic areas. Mechanistically, under hypoxic conditions, ROS-dependent induction of IL4 has been shown to contribute to the recruitment and M2 polarization of macrophages. TAMs preferentially localize in these hypoxic areas of a tumor [37]. Lactate, produced by tumor cells via anaerobic glycolysis, enhances macrophage ROS production and stimulates Nrf2 activation [102]. Furthermore, M2-derived vascular endothelial growth factor (VEGF) leads to the activation of NRF2 in neighboring tumor cells to support epithelial-to-mesenchymal transition (EMT) [102]. The presence of TAMs reduces the effectiveness of tumor therapy protocols, which applies to both chemotherapy and radiotherapy as well as angiogenesis inhibition [100,103,104]. Activation of NRF2 in cancer cells—via induction of target gene expression—promotes cancer progression, metastasis, and resistance to radio- or chemotherapy [98].

By applying the CSC model, sphere culture of HT-116 colon cancer cells, it could be demonstrated that the expression of the colon cancer CSC marker CD133 correlates with that of Nrf2. Silencing of CD133 reduced the expression of stem cell markers Kruppel-like factor 4 (KLF4) and ABCC2, but decreased the expression of NRF2 [105]. Mechanistically, this effect seems to be mediated via PI3K/GSK-3β. Vice versa, silencing of *NRF2* impaired sphere formation and led to a decrease in CSC markers [105]. Furthermore, in the CD133^high^ population isolated from HT-116 cells, it was shown that CSCs exhibit higher expression of NRF2, and this is associated with enhanced CSC-like properties such as the extent of sphere formation, colony formation, and resistance to anticancer therapy. The authors conclude that the CD133/NRF2 axis contributes to the development of a CSC-like phenotype [105]. High expression of the antioxidant gene peroxiredoxin-2 (*PRDX2*) has been attributed to CD133^+^CD44^+^ cells and is associated with colon cancer CSC stemness [106]. Depletion of *PRDX2* substantially suppressed maintenance of stemness, together with migration, invasion, and metastasis. Notably, knockdown of PRDX2 increased ROS-levels in CD133^+^CD44^+^, making CSCs sensitive to oxidative stress and chemotherapy [106].

An association between CSC markers and Nrf2 has been detected in several studies. The CD44^high^CD24^low^ CSC population exhibited higher expression of CSC makers and of Nrf2. Nrf2 levels could be further increased by addition to the cells of hyaluronic acid, a ligand for CD44 [107]. Accordingly, high expression levels of CD133 and CD44 go along with increased NRF2 and its target gene, efflux transporters ABCC1 [93]. In ovarian CSCs, Nrf2 levels were found to be associated with levels of ALDH [89].

## 6. Signaling Pathways Contributing in the Regulation of ROS Levels in CSC

Different signaling pathways support and enable the self-renewal of cancer stem cells. These include the Wingless-Int (Wnt), Notch, and Sonic Hedgehog (HH) pathways [2,108,109]. Notably, all these pathways to some extent contribute to CSC maintenance by regulating CSCs’ redox balance.

### 6.1. Wnt-Signaling Pathway

Hyperactivation of the Wnt/β-catenin signaling pathway has been identified as one of the most frequent events occurring in CSCs [110]. Activation of the pathway leads to stabilization and nuclear translocation of β-catenin and, eventually, transcriptional upregulation of target genes [111,112]. Notably, the Wnt/β-catenin pathway has been heavily implicated in liver CSCs [113,114]. In hepatic CSCs, glutamine synthase 1 (GLS1) is highly expressed and regulates antioxidant defense function by increasing GSH levels and decreasing reactive oxygen species (ROS) levels, which in turn protects cells from oxidative stress. Silencing *GLS1* expression or inhibiting GLS1 activity dysregulated the redox homeostasis of cancer cells [115,116,117]. GLS1 expression is positively associated with the stemness phenotype in hepatocellular carcinoma (HCC), and targeting GLS1 inhibits CSC markers expression and stem-like properties in vitro and in vivo. Mechanistically, glutamine deprivation or GLS1 inhibition leads to an increased accumulation of ROS, which suppresses the translocation of β-catenin from the cytoplasm to the nucleus and consequently decreases the expression of stemness-related genes [118]. Similarly, glutamine has been shown to support the maintenance of stemness by promoting glutathione synthesis. Accordingly, deprivation of glutamine in the culture medium reduced the proportion of side population cells of the non-small lung cancer A549 cell line [119]. Deprivation of glutamine, similar to the addition of H_2_O_2_, increased the phosphorylation of β-catenin [111], the central player in the canonical Wnt-pathway [111,112], and, thus, downregulated *SOX2* and other Wnt-regulated genes such as baculoviral IAP repeat containing 5 (*BIRC5*, survivin) and *AXIN1* [118]. Inactivation of the adenomatous polyposis coli (*APC*) gene is frequently observed in colorectal carcinoma. It is considered the initiating event in colorectal tumorigenesis [120]. Inactive APC mutants exhibit constitutive activation of the β-catenin signaling [111]. APC is a negative regulator of the canonical Wnt-pathway, which provides a self-renewal signal for different types of stem cells [121]. Mechanistically, LGR5^+^/intestinal stem cell expansion after loss of *APC* was dependent on RAC1 and ROS production, as well as on efficient NF-κB (nuclear factor ‘kappa-light-chain-enhancer’) of activated B-cells (NF-κB)—signaling [122].

Irradiation injury induces proliferation of an intestinal stem cell subset [123]. This process was shown to be mediated by upregulation of ROS production in the crypts, concomitant induction of HIF-1α, which, among others, directly transactivated *WNT2B* leading to activation of β-catenin [123].

The non-canonical branches of the Wnt-signaling pathway have also been implicated in stem cell quiescence [124]. WNT5A was able to inhibit ROS production of hematopoietic stem cells via the receptor-like tyrosine kinase (RYK) [124].

### 6.2. Sonic Hedgehog Signaling

The Sonic Hedgehog signaling pathway is crucially implicated in carcinogenesis and the tumor microenvironment. Aberrant HH signaling activation accelerates tumor growth and, as far as CSCs are involved, contributes to immune tolerance and drug resistance [125,126]. In gastrointestinal CSCs, various factors lead to the activation of HH signaling and, thereby, increase stemness. These factors include chemotherapy, CCN1 and NOTCH1 [127], SKI [128], and vasohibin 2 [129], which upregulate stem cell markers such as CD24, CD44, CD133, SOX2, OCT4, and NANOG as well as efflux transporters ABCC1 and ABCG2, and hypoxia-induced factor 1α (HIF1α) [130]. The latter is a key transcription factor, which is crucial for adapting cells and tissues to diminished oxygen availability [131] by stimulating oxygen supply via increasing angiogenesis and erythropoiesis and limiting oxygen consumption via shifting energy metabolism from oxidative phosphorylation to glycolysis [132]. This metabolic switch is considered to contribute to low ROS levels as they are required for maintaining stemness [133]. Activation of the HH pathway has been shown to stimulate transcription of the glycolytic gate keepers hexokinase 2 (HK2) and pyruvate kinase 2 (PKM2) [134]. Transcriptional activation of these genes is mediated by activation of GLI transcription factors, which are central to the canonical HH pathway.

Positive expression of GLI family zinc finger 1 (GLI1) has been recognized as a reliable indicator of poor prognosis in patients with highly aggressive gastric cancer [135]. The poor prognosis of patients with strong expression of GLI1 very likely is due to reactive oxygen species (ROS) generated by NADPH oxidase 4 (NOX4). Of note, increased production of ROS in hypoxia leads to GLI1 induction and EMT [136,137]. Interestingly, in human colon cancer biopsies, overexpression of the HH pathway components sonic hedgehog (SHH), patched 1 (PTCH), or GLI1, was found to be an indicator of poor prognosis [125]. Likewise, overexpression of the frizzled class receptor smoothend (SMO) has been suggested to be associated with poor prognosis in colorectal cancer [138,139].

Tumor-associated hypoxia contributes both to drug resistance and self-renewal. Inhibition of HH signaling was reported to suppress self-renewal and chemo-resistance of, e.g., pancreatic CSCs [140]. GLI target genes regulate not only proliferation, migration/invasion, and angiogenesis, but also stem cell regeneration (via target genes FST1, BMP4) [141]. EMT has been demonstrated to endow cells with stem cell-like properties. Accordingly, cananginone, a secondary metabolite of certain plants, was shown to inhibit HH signaling, EMT, and the acquisition of stem cell features in parallel. Inhibition of HH, and of GLI1 in particular, mediated this induction of EMT, largely by interfering with transcriptional activation of, e.g., SNAIL and VIM [142]. Likewise, engrailed (EN1), a homeo-box containing transcription factor that regulates embryonic development, has been shown to control tumorigenic capacity and resistance to radiation of glioma by regulating intracellular ROS levels [143]. Again, regulation of HH by altering GLI1 expression level has been identified as the underlying mechanism.

ROS-induced activation of NRF2 controls drug resistance, for example against sorafenib [144]. Mechanistically, the authors could demonstrate that NRF2 physically binds to the promoter of Sonic Hedgehog Signaling Molecule (SHH) [144], and that the administration of recombinant SHH abolished the effects resulting from NRF2 knockdown [144].

Of note, there are other cellular signaling pathways that interact with HH signaling. This holds true, e.g., for the Wnt pathway, transforming growth factor/bone morphogenic protein (TGF/BMP) pathway, and the epidermal growth factor (EGF) receptor pathway [145]. As far as the Wnt pathway is concerned, in the developing nervous system, upstream HH activation is absolutely required for the expression of T cell factors 3/4 (TCF3/4), major transcriptional mediators of the canonical Wnt signaling pathway [145].

### 6.3. CD13/APN in CSC

A positive correlation between multidrug resistance and CD13 expression has been established [146,147,148]. A set of surface markers representative of liver cancer stem cell (LCSC) subpopulations has been identified in in hepatocellular carcinoma (HCC). These include CD133, CD44, CD13, CD90, CD47, epithelial cell adhesion molecule (EpCAM), an antigen detected by “oval cell” antibody, OV-6, and the side population. CD13^+^ cells were mainly detected in the G0/G1 phase. They comprised dormant or slow-growing cancer cell populations, which are associated with chemoresistance due to ABC transporter expression and recurrence [149]. Similar to CD44, CD13 in combination with other surface markers, including CD133 or CD90, could effectively initiate tumor formation, as observed by increased HCC tumorigenesis [63]. CD13^+^CD133^+^ and CD13^+^CD90^+^ cells increase HCC tumor initiation and genotoxic chemoresistance to, e.g., doxorubicin (DXR) and fluorouracil (5′Fu) in parallel [150]. Accordingly, expression of hepatic stem cell markers (CD13, CD90, CD133), the proportion of side population, ability to induce sphere formation, and in vivo tumorigenesis were highest in selected HCC cells resistant to doxorubicin or 5′-fluorouracil [151]. Recent data suggest that CD13^+^ CSCs depend on aerobic metabolism of tyrosine rather than glucose as an energy source [143]. The acetyl-CoA formed by the metabolism of tyrosine facilitates acetylation/stabilization of transcription factor FOXD3, which in turn contributes to quiescence and chemoresistance [152].

CD13 (APN, EC3.4.11.2) is a Zn^2+^-dependent ectopeptidase that preferentially cleaves small neutral amino acids from the N-terminus of small peptides. APN/CD13 is associated with diverse physiological functions, including proliferation, differentiation, invasion, migration, and angiogenesis [153]. Increased expression or activity of APN/CD13 has been described for a variety of different tumors, with aggressive and rapidly growing tumors often being strongly CD13-positive (CD13^+^). For colon cancer, expression of CD13 has been shown to be associated with reduced disease-free and overall survival [154].

APN/CD13 is a neo-angiogenesis marker that is found on neo-angiogenetically active, but not on normal, endothelia [155]. This fact makes APN/CD13 an interesting and specific receptor for so-called “tumor-homing peptides” (NRG peptides), which bind to APN with high affinity, and thus represent a promising target for the inhibition of tumor-induced angiogenesis as well as for the tumor-selective administration of, e.g., cytotoxic substances [156,157,158]. In this way, APN/CD13 can, in principle, also be used for molecular imaging of tumor masses and (neo)angiogenesis in animal models and also in patients [159]. Pharmacological inhibition of the enzyme has been shown to reduce tumor growth and progression in a number of APN/CD13-positive tumors. The inhibition of angiogenesis (or “tube formation” in vitro) is only one obvious aspect, because therapeutic effects of APN inhibitors (ubenimex, actinonin) have also been described for acute or chronic myeloid leukemia (AML, CML) [160,161,162].

CD13 has been identified as a CSC-specific membrane marker (e.g., for hepatocellular carcinoma, cholangiocarcinoma) [163,164]. CD13 is preferentially expressed on cells of the side population (SP), which are characterized by extremely high chemoresistance and tumorigenicity in a transplantation model [165,166]. Functionally, CD13 is known to reduce the extent of DNA damage induced by ROS production after genotoxic stress and thereby protect CSC from apoptosis [165]. In hepatic cancer cells, the TGF-β-induced epithelial-to-mesenchymal transition (EMT) is associated with increased expression of CD13, which prevents a further increase in ROS levels and, thus, supports CSCs’ survival [64]. The pharmacological inhibition of APN/CD13 stimulates CSCs apoptosis [64] and has been shown to increase the antitumor effects of chemotherapeutic agents, e.g., 5-fluorouracil [148,167], which is why APN/CD13 inhibitors are considered *cancer chemosensitizers* [168]. A sensitization of the CSC to chemotherapeutic agents has also been achieved by the administration of APN/CD13 inhibitors in colorectal and hepatocellular carcinoma [150,169].

CD13 expression was found to be higher in metastatic HCC samples, and was associated with poor prognosis for patients after surgical resection [170]. CD13 stimulated HCC proliferation, invasion, and progression through the cell cycle, and caused resistance to sorafenib. Mechanistically, this was achieved by an activation of nuclear factor kappa B (NF-κB)-signaling, which was initiated by a direct interaction with and stabilization of histone deacetylase 5 (HDAC5), leading to lysin-specific demethylase 1 (LSD1) deacetylation/stabilization [170].

Available data suggest that CD13 induces multi-drug resistance (MDR) in tumor cells [148]. MDR is a phenomenon that makes certain cells resistant to the action of chemotherapy drugs, and is regarded as a characteristic stem cell feature. MDR often is achieved through increased expression of efflux transporters. In CSCs, the induction of ABC transporter expression via CD13 is caused by dysregulated HH signaling [171], where CD13 can act as a pseudoligand of the receptor PATCHED to sensitize the pathway, and to upregulate expression of ABCG2, ABCB1 (P-glycoprotein), ABCC2 (MRP2), and ABCC3 (MRP3) to increase drug resistance [172].

There is a negative correlation between CD13 expression and the cellular level of ROS [173]. In contrast, CD13 is positively correlated with expression of anti-apoptotic BCL2 and BCL-XL [174]. Mechanistically, ROS as second messenger and mitogen-activated protein kinases (MAPK) are involved in regulation of drug resistance. Of note, inhibition of APN/CD13 has been demonstrated to provoke induction/activation of mitogen-activated protein kinases (MAPK) Erk1/2 in the human T cell line KARPAS-299 [175].

In HCC, dormant CD13^+^ CSCs residing in hypoxic areas of the tumor survive radiation or chemotherapy [63,176]. These dormant CSCs exhibit significantly less ROS-induced double-stranded breaks (DSB) than CD13^-^ cells, which undergo ROS-mediated apoptotic cell death following chemotherapy [177].

## 7. Conclusions

The CSC phenotype is characterized by a high self-renewing capability and resistance to chemotherapy drugs, among other things. Its maintenance is crucially dependent on ensuring low intracellular ROS levels and induced expression of efflux transporters. Low ROS levels are largely due to the increased expression of anti-oxidative structures/mechanisms. The NRF2 plays an important role in this. Furthermore, metabolic balancing between anaerobic glycolysis and oxidative phosphorylation minimizes ROS production. Signaling pathways and molecules associated with the CSC phenotype, such as the Wnt- and HH pathways, and inhibition of APN/CD13 enzymatic activity regulate the maintenance of the CSC phenotype in several ways. Direct or indirect regulation of the CSC redox balance is crucially implicated in CSC maintenance.

Pharmacological targeting of the signaling pathways involved can represent a promising strategy for overcoming the limitations of current tumor therapy protocols. APN/CD13 facilitates the targeted inhibition of tumor angiogenesis and the sensitization of CSC for radiotherapy and chemotherapy. Likewise, HH (SMO and GLI) inhibitors are promising tools to address stemness-associated features of CSC. The contribution of ROS in these pathways requires further investigation.

## Figures and Tables

**Figure 1 biomedicines-10-02413-f001:**
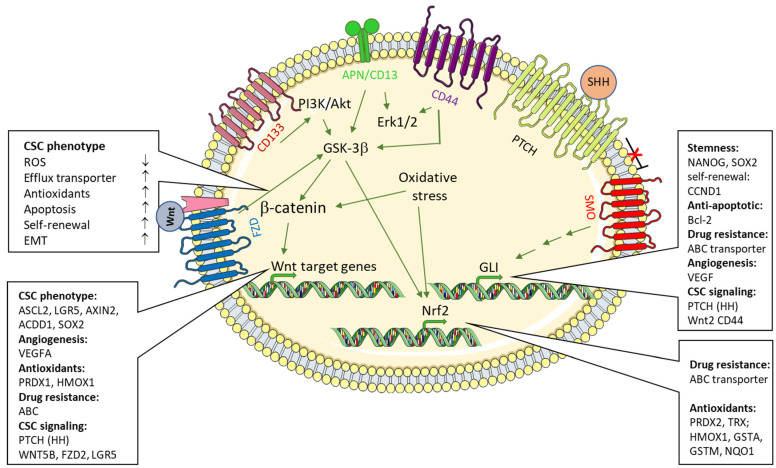
Pathways contributing to the maintenance of the cancer stem cell (CSC) phenotype. Keeping intracellular ROS levels low is required for a stem cell to maintain quiescence and self-renewal properties. This is achieved to a large extent by expression of Nrf2-induced target genes. Thereby, antioxidative defense molecules and detoxifying enzymes including thioredoxin, peroxiredoxin, glutathione-S-transferases, heme oxygenase I are provided to keep the levels of reactive oxygen species derived from, e.g., NADH oxidases (NOX), the respiratory chain, infiltrating/surrounding immune cells, or chemotherapeutic drugs low. Nrf2 and pathways initiated by the CSC surface receptors and markers to minimize ROS production in response to chemotherapeutics like Sonic Hedgehog, Wnt family members, CD13, CD44 or CD133 add to the strong induction of drug efflux transporters. In addition, pathways that contribute to the CSC phenotype act via preventing apoptosis, promoting angiogenesis and expression of stem cell transcription factors such as NANOG homeobox (NANOG HOMEOBOX) and SRY-box transcription factor 2 (SOX2). Note that the canonical β-catenin-dependent Wnt pathway, and GSK-3β in particular, is a downstream mediator shared by many of the CSC markers and CSC pathways.

**Figure 2 biomedicines-10-02413-f002:**
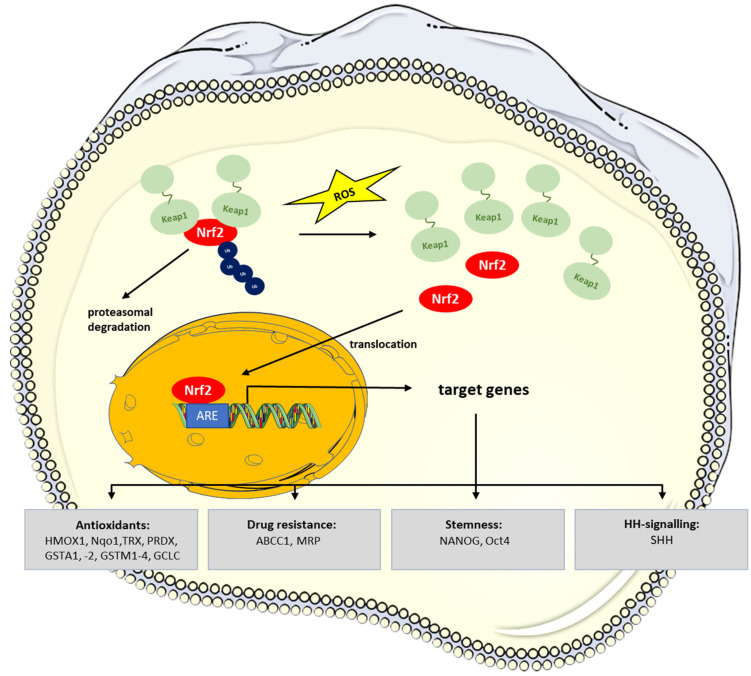
Nrf2 signaling pathway activation by ROS and induction of target genes. Under physiological conditions, NRF2 bound to KEAP1 is ubiquitylated and degraded in the proteasome. In response to ROS and various other stress factors, KEAP1 is released from NRF2, leading to its stabilization and translocation into the nucleus. Target gene expression is induced upon binding of NRF2 to AREs (antioxidant response element) in the promoters of target genes. Target genes comprise genes coding for the cell’s antioxidative equipment, efflux pumps enabling drug resistance, stemness-regulating transcription factors, as well as SHH (modified from [71,72]).

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
