# Peer review of "Redox-Regulation in Cancer Stem Cells"

_biomedicines, 2022, doi:10.3390/biomedicines10102413_

Round 1

Reviewer 1 Report

In this review paper, the authors described maintenance of stemness and drug resistance of cancer stem cells by various antioxidant proteins in the cancer stem cells. This work is important because it highlights a vulnerability in CSCs which may be resistant to cancer therapies. Overall I find the design and direction excellent. However, it feels a bit out of order. For example, when describing the role of NRF2 as a transcription factor, it is necessary to have an organized figure or a systematic description of the target gene (Antioxid Redox Signal. 2018 Dec 10; 29(17):1727–1745).

Also there are some grammatical errors. However, I do consider that the holes can easily be repaired and much of the fundamentals of the paper are solid.

Author Response

We would like to thank the reviewers and editors for a thorough review of our manuscript. With the help of the very helpful comments and advice, we believe that the manuscript has been significantly improved. Thank you very much!

Our response in detail to the points raised by the reviewers follows here:

In this review paper, the authors described maintenance of stemness and drug resistance of cancer stem cells by various antioxidant proteins in the cancer stem cells. This work is important because it highlights a vulnerability in CSCs which may be resistant to cancer therapies. Overall I find the design and direction excellent. However, it feels a bit out of order. For example, when describing the role of NRF2 as a transcription factor, it is necessary to have an organized figure or a systematic description of the target gene (Antioxid Redox Signal. 2018 Dec 10; 29(17):1727–1745).

We thank the reviewer for this comment. The structure of the text has been revised and we include a new Figure 2 to better introduce into the important Nrf2 pathway.

Also there are some grammatical errors. However, I do consider that the holes can easily be repaired and much of the fundamentals of the paper are solid.

We thank the reviewer for pointing this weakness to us. We have tried to find and correct all errors in the text.

Reviewer 2 Report

This review article by Lendeckel and Wolke is aimed at reviewing bibliography on the topic of Redox regulation and cancer stemness. Unfortunately, even though the review is nicely written and starts really good, for some reason authors lost the focus in the middle of the article and start reviewing articles that have nothing to do with CSC or ROS regulation. What definitely will make readers loose interest in this work.

An overall comment: use of acronyms is loose. Some acronyms such as ROS or SP are defined twice, while many are not defined ever, or not the first time they are mentioned. For example, WNT is mentioned in page 1, and its acronym defined in page 6. Please make sure you define all acronyms, and you do it in the right place.

Also, in all the review the use of capital for proteins, or italic for genes is messy, and I do not feel authors follow any rule.

Page 1, line 20: I would say well established and not “best established”, as people might have a different opinion regarding which is the “best” here.

Page 2, second paragraph, line 3: remove “in vivo”. If you say transferred in mice… everyone will understand this is in vivo.

Page 3, line 7: “A marker-free population of glioma-inducing…” This sentence does not flow in the text, and topic is not well explained. Either authors explain it better or remove it.

Page 3, paragraph 2, line 2: change the order to “cell signaling, therapy resistance and tumor recurrence” please.

Page 3, paragraph 3, line 5 to 8. This few sentences are not explained right. It makes me think that SOD induces ROS production, while it is an antioxidant enzyme. This should be re-written and better explained.

Page 4, line 4:  CD133 and ALDH1 are not transcription factors!

Page 4, paragraph 2, line 10: “The strong induction, however, in tumors…” This sentence makes no sense the way is written.

Page 4, paragraph 2, line 16: “…and the capacity of the defense system” to block them?

Page 4, paragraph 2: Authors have a big paragraph just to explain what ROS and oxidative stress are. I feel this could be far shorter (3-4 lines) and give same information.

Page 4, paragraph 3: CSC are said here to have low ROS levels same as normal stem cells, and less than non-CSC. Non-CSC is tumor bulk? It is confusing the use of Non-CSC.

Page 4, paragraph 3: what is snail? If it is a protein should be capitalized, I don’t think authors mean the animal here. They should also write a couple of words to define Snail. Something like transcriptional repressor involved in EMT….maybe.

Page 4, paragraph 4: “Later this could be confirmed in CD13+ liver CSC” in which ….. explain what happen in these cells.

Page 5, paragraph 2, line 3: the use of “detrimental” is confusing here. Does it mean reduces or increases proliferation, and is it good of bad for resistance?. Please re-write this sentence.

Page 5, paragraph 5, line 1: what does it mean “can induce CSC”? de-differentiate maybe?

Page 5, paragraph 6, line 3: “sonic hedgehog” should be Sonic Hedgehog unless talking about the video game. Also do the authors mean signaling or ligand here? The rest of the manuscript authors say HH, not SHH specifically.

Page 5, paragraph 8: this paragraph has nothing to do with CSC, and it should be cut up.

Page 6, paragraph 2, line 3: fix the sentence to “… silencing of CD133 reduced the expression of not only stem cell markers KLF4 and ABCC2, but also decreased the expression of Nrf2”

Page 6, paragraph 2, line 6: “In another CSC model…”. It is same model, but different manuscript. Both references are HT-166 CRC cells and mention CD133 cells.

Page 6, paragraph 3, line 1: Authors should not start a new paragraph by saying “Furthermore”. In this same line associated with? Instead of “in”.

Page 6, paragraph 3, line 2: What is CSC behavior? Also, CSC has been defined before… HH and WNT were mentioned before, and here they are defined.

Page 6, paragraph 3, line 7-12: 7 lines talking about CSC, GSL1 and ROS…. But no link with WNT, which is the topic of this section.

Page 6, paragraph 3, line 15-18: Similarly, no link with WNT.

Page 6, paragraph 3, line 20: B-catenin was mentioned a few lines before, but now authors describe its key role in WNT signaling. It should be mentioned before.

Page 7, paragraph 2: I do not see the need of using an acronym for IR here, as I do not think it is mentioned after this line anymore.

Page 7, paragraph 4: The complete HH section should be re-written. Only the last four lines in this section have something to do with ROS. Also the tittle should be Hedgehog signaling, and the acronym defined in the first place in the manuscript where HH was mentioned, not here. Also, again hedgehog is an animal, Hedgehog is the signaling pathway.

Page 7-8, section of CD13/APN: This section should have been named CD13/APN in CSC or CD13/APN- regulation of ROS levels in CSC… something similar. But they cannot use a title like this, because most of this section has nothing to do with ROS or CSC. Four complete and long paragraphs with no link with ROS or CSC at all, in a review of ROS and CSC. Complete loss of focus. ROS and CSC are not mentioned until paragraph 5. Those 4 paragraphs should be reduced to one.

Page 8, paragraph 6: Define MDR, and explain what is it. Although again this paragraph has nothing to do with ROS – the topic of this review.

Page 9: authors should write a conclusion paragraph at the end of the review.

Author Response

We would like to thank the reviewers and editors for a thorough review of our manuscript. With the help of the very helpful comments and advice, we believe that the manuscript has been significantly improved. Thank you very much!

Our response in detail to the points raised by the reviewers follows here:

This review article by Lendeckel and Wolke is aimed at reviewing bibliography on the topic of Redox regulation and cancer stemness. Unfortunately, even though the review is nicely written and starts really good, for some reason authors lost the focus in the middle of the article and start reviewing articles that have nothing to do with CSC or ROS regulation. What definitely will make readers loose interest in this work.

We thank you for the comment and have subsequently restructured the manuscript and questioned all chapters in relation to ROS.

 An overall comment: use of acronyms is loose. Some acronyms such as ROS or SP are defined twice, while many are not defined ever, or not the first time they are mentioned. For example, WNT is mentioned in page 1, and its acronym defined in page 6. Please make sure you define all acronyms, and you do it in the right place.

We thank the reviewer for pointing this inconsistency to us. Changes have been made accordingly.

 Also, in all the review the use of capital for proteins, or italic for genes is messy, and I do not feel authors follow any rule.

We apologize for the messy use of gene and protein names. We made the necessary corrections. only when referring to the pathways (e.g. Wnt- and hedgehog pathway) we have deviated from it.

Page 1, line 20: I would say well established and not “best established”, as people might have a different opinion regarding which is the “best” here.

Done.

Page 2, second paragraph, line 3: remove “in vivo”. If you say transferred in mice… everyone will understand this is in vivo.

Done.

Page 3, line 7: “A marker-free population of glioma-inducing…” This sentence does not flow in the text, and topic is not well explained. Either authors explain it better or remove it.

The text has been changed.

 Page 3, paragraph 2, line 2: change the order to “cell signaling, therapy resistance and tumor recurrence” please.

Done.

 Page 3, paragraph 3, line 5 to 8. This few sentences are not explained right. It makes me think that SOD induces ROS production, while it is an antioxidant enzyme. This should be re-written and better explained.

Here we had trouble recognizing the problem, but have changed the text, but hopefully satisfactorily.

 Page 4, line 4:  CD133 and ALDH1 are not transcription factors!

We thank the reviewer for pointing this error to us. It has been corrected.

Page 4, paragraph 2, line 10: “The strong induction, however, in tumors…” This sentence makes no sense the way is written.

Thank you! We have changed the text.

 Page 4, paragraph 2, line 16: “…and the capacity of the defense system” to block them?

Yes, in a way to block them. We changed the text to better explain the fact.

 Page 4, paragraph 2: Authors have a big paragraph just to explain what ROS and oxidative stress are. I feel this could be far shorter (3-4 lines) and give same information.

Here we had the problem of not knowing what the other articles of the special issue do contain. We consider the declaration to be very important. Nevertheless, we tried to cut back.

 Page 4, paragraph 3: CSC are said here to have low ROS levels same as normal stem cells, and less than non-CSC. Non-CSC is tumor bulk? It is confusing the use of Non-CSC.

We are grateful to the reviewer for pointing this lack of clarity to us. Yes, tumor bulk is a good expression. It should be distinguished from the CSC. We have changed the text accordingly.

 Page 4, paragraph 3: what is snail? If it is a protein should be capitalized, I don’t think authors mean the animal here. They should also write a couple of words to define Snail. Something like transcriptional repressor involved in EMT….maybe.

Again, thanks for pointing this lack of clarity to us. Changes have been made accordingly.

 Page 4, paragraph 4: “Later this could be confirmed in CD13+ liver CSC” in which ….. explain what happen in these cells.

Done.

 Page 5, paragraph 2, line 3: the use of “detrimental” is confusing here. Does it mean reduces or increases proliferation, and is it good of bad for resistance?. Please re-write this sentence.

Thanks! Done.

 Page 5, paragraph 5, line 1: what does it mean “can induce CSC”? de-differentiate maybe?

We would like to thank the reviewer for this comment. We changed the text for better understanding.

 Page 5, paragraph 6, line 3: “sonic hedgehog” should be Sonic Hedgehog unless talking about the video game. Also do the authors mean signaling or ligand here? The rest of the manuscript authors say HH, not SHH specifically.

We are grateful for pointing this inconsistency to us. Where we refer to the ligand, we write now “sonic hedgehog (SHH)”. When referring to the pathway we write “HH”.

  Page 5, paragraph 8: this paragraph has nothing to do with CSC, and it should be cut up.

Here we would like to ask the reviewer for leniency. We see some connection in the mechanisms to the facts explained in other sections (NRF2 induction, therapy resistance, EMT, etc.), which perhaps makes this section interesting in context. It also sensitizes the reader to the role of M1 versus M2 macrophages, which is in any case insufficiently studied.

 Page 6, paragraph 2, line 3: fix the sentence to “… silencing of CD133 reduced the expression of not only stem cell markers KLF4 and ABCC2, but also decreased the expression of Nrf2”

Done.

 Page 6, paragraph 2, line 6: “In another CSC model…”. It is same model, but different manuscript. Both references are HT-166 CRC cells and mention CD133 cells.

We are very grateful for this helpful comment. Changes have been made accordingly.

 Page 6, paragraph 3, line 1: Authors should not start a new paragraph by saying “Furthermore”. In this same line associated with? Instead of “in”.

Corrected.

 Page 6, paragraph 3, line 2: What is CSC behavior? Also, CSC has been defined before… HH and WNT were mentioned before, and here they are defined.

We thank the reviewer for pointing this lack of clarity to us. We wanted to refer to self-renewal itself and omitted the term "behavior" in the revised text.

 Page 6, paragraph 3, line 7-12: 7 lines talking about CSC, GSL1 and ROS…. But no link with WNT, which is the topic of this section.

The Wnt pathway is inextricably intertwined with CSC, most certainly via ROS, at least. The chapter should not be deleted. The section lists 9 references that directly relate to it. Further passages introduce e.g. APC or GSL1, underlaid with further references.

For better visibility, we have highlighted the directly Wnt-relevant lines in yellow.

Page 6, paragraph 3, line 15-18: Similarly, no link with WNT.

See above.

Page 6, paragraph 3, line 20: B-catenin was mentioned a few lines before, but now authors describe its key role in WNT signaling. It should be mentioned before.

We are grateful for this comment. The text has been changed accordingly.

Page 7, paragraph 2: I do not see the need of using an acronym for IR here, as I do not think it is mentioned after this line anymore.

“(IR)” has been omitted from the text.

Page 7, paragraph 4: The complete HH section should be re-written. Only the last four lines in this section have something to do with ROS. Also the tittle should be Hedgehog signaling, and the acronym defined in the first place in the manuscript where HH was mentioned, not here. Also, again hedgehog is an animal, Hedgehog is the signaling pathway.

Again, there is much relation to mechanisms addressed in other chapters even if ROS is not directly mentioned (e.g. with NRF2). So, we do not agree 100 % with the reviewer. Suggested changes have been made, anyway, to the text and title.

 Page 7-8, section of CD13/APN: This section should have been named CD13/APN in CSC or CD13/APN- regulation of ROS levels in CSC… something similar. But they cannot use a title like this, because most of this section has nothing to do with ROS or CSC. Four complete and long paragraphs with no link with ROS or CSC at all, in a review of ROS and CSC. Complete loss of focus. ROS and CSC are not mentioned until paragraph 5. Those 4 paragraphs should be reduced to one.

Thanks for this comment.

The section has been renamed as suggested. APN/CD13 has been successfully applied for tumor therapy and was shown to target CSCs (e.g. in colorectal cancer or HCC). It is a clinically relevant option that is in sight for few other pathways or structures, maybe down-stream HH antagonists are “drugable”. That alone justifies the place in the manuscript. Inhibition of APN (by Ubenimex or similar compounds) sensitizes CSC for chemotherapy, very likely ROS or the lost protection against ROS is involved here (and we consider this as the final goal of cancer therapy), but also inhibits selectively tumor angiogenesis and facilitates specific drug-targeting of tumors. We are convinced this is worth mentioning, although we might be biased here.

APN/CD13 is proven to be associated with WNT-signaling and ROS. The chapter on APN/CD13 must not be not be viewed and evaluated on its own but in context.

As we see the point the reviewer makes, we have reduced Wnt-, HH-signaling and APN/CD13 to one chapter.

Page 8, paragraph 6: Define MDR, and explain what is it. Although again this paragraph has nothing to do with ROS – the topic of this review.

Done.

 Page 9: authors should write a conclusion paragraph at the end of the review.

We thank the reviewer for this helpful advice. Done.

Round 2

Reviewer 2 Report

The manuscript by Lendeckel and Wolke has been improved after initial review. 

I would just add that the conclusion section is little long for a mini-review article. It should be just 4-5 lines. Also in these lines SHH, HH or hedgehog signaling words are used randomly. Please make sure you use or not acronyms all times, and use the same. 

Unless authors have a problem with word count, I would recommend them to use the minimal amount of acronyms, and only for those words heavily repeated. Acronyms for diseases should be avoided to help reader follow the text.

The section of Side population doesn't seem to be linked to OS, and therefore not sure why it is included in this review. Could authors find a way to make the section fit with the topic of the review?

Author Response

We would like to thank the reviewer for the thorough review of our manuscript, which helped a lot to improve its quality.

The manuscript by Lendeckel and Wolke has been improved after initial review.

We thank the reviewer for this positive comment on our manuscript.

I would just add that the conclusion section is little long for a mini-review article. It should be just 4-5 lines. Also in these lines SHH, HH or hedgehog signaling words are used randomly. Please make sure you use or not acronyms all times, and use the same.

Thanks for this comment. We have shortened the conclusion section by 27 % (from 242 to 177 words) and use now HH for the sonic hedgehog pathway and SHH for the ligand, sonic hedgehog.

Unless authors have a problem with word count, I would recommend them to use the minimal amount of acronyms, and only for those words heavily repeated. Acronyms for diseases should be avoided to help reader follow the text.

We thank the reviewer for this advice.

The section of Side population doesn't seem to be linked to OS, and therefore not sure why it is included in this review. Could authors find a way to make the section fit with the topic of the review?

We are grateful to the reviewer for this comment. We have rewritten the section and added references that indeed studied ROS in SP. We also tried to make more clear that SP cells are indeed stem cells or even CSCs and that all stem-like features we elaborate on in other sections (low ROS levels, high antioxidant equipment, self-renewal, and high efflux activity apply to SP cells. SP cells represent just one (universal but very important) way to sort / identify these cells.
